# Genomic characterization of DICER1-associated neoplasms uncovers molecular classes

Felix K. F. Kommoss [1], Anne-Sophie Chong [2,3,4], Anne-Laure Chong[2,3,5], Elke Pfaff[6,7,8], David T. W. Jones [6,7], Laura S. Hiemcke-Jiwa[9,10], Lennart A. Kester [10], Uta Flucke[10,11], Manfred Gessler [12], Daniel Schrimpf[13,14], Felix Sahm [13,14], Blaise A. Clarke[15], Colin J. R. Stewart[16,17], Yemin Wang [18,19], C. Blake Gilks[18], Friedrich Kommoss[20], David G. Huntsman[18,19], Ulrich Schüller [21,22,23], Christian Koelsche [1], W. Glenn McCluggage[24], Andreas von Deimling [13,14,25] & William D. Foulkes [2,3,5,25] ✉

DICER1 syndrome is a tumor predisposition syndrome that is associated with up to 30 different neoplastic lesions, usually affecting children and adolescents. Here we identify a group of mesenchymal tumors which is highly associated with DICER1 syndrome, and molecularly distinct from other DICER1-associated tumors. This group of DICER1-associated mesenchymal tumors encompasses multiple well-established clinicopathological tumor entities and can be further divided into three clinically meaningful classes designated "low-grade mesenchymal tumor with DICER1 alteration" (LGMT DICER1), "sarcoma with DICER1 alteration" (SARC DICER1), and primary intracranial sarcoma with DICER1 alteration (PIS DICER1). Our study not only provides a combined approach to classify DICER1-associated neoplasms for improved clinical management but also suggests a role for global hypomethylation and other recurrent molecular events in sarcomatous differentiation in mesenchymal tumors with DICER1 alteration. Our results will facilitate future investigations into prognostication and therapeutic approaches for affected patients.

DICER1 is a cytoplasmic endoribonuclease that is critical for the correct processing (cleavage) of precursor micro-RNA (pre-miRNA) double-stranded hairpins with 3' and 5' ends to their mature single-stranded forms[1,2]. DICER1 utilizes its RNAse IIIa and IIIb domains to process pre-miRNAs, yielding a duplex containing either a mature 5p or 3p miRNA as well as a complementary 'passenger strand' that is ultimately degraded. The mature miRNA is then loaded onto an AGO protein to form an RNA-induced silencing complex, eventually resulting in down-regulation or silencing of mRNA targets. Specific metal-ion binding amino acids within the IIIa and IIIb domains are crucial for pre-miRNA cleavage. Failure of dicing of the pre-miRNA by

RNase IIIb is a critical event in most DICER1-associated tumors. This failure arises because of the occurrence of tumor-confined missense variants that result in amino acid substitutions at these critical pre-miRNA-interacting residues within RNase IIIb[3,4].

DICER1 syndrome is a tumor susceptibility syndrome, characterized in 2009, determined by the occurrence of a germline pathogenic variant (PV) in DICER1[5]. Typically, this is a loss of function (LOF) variant that is predicted to result in inactivation of the affected DICER1 allele. For the syndrome to occur, a second hit affecting exons encoding the RNase IIIb domain of DICER1 (a "hotspot" PV), as discussed above, is usually required[3,4]. The phenotypic spectrum of DICER1 syndrome is

wide but some of the manifestations almost exclusively occur in persons with DICER1 syndrome. Both benign and malignant neoplasms are part of the syndrome. Pleuropulmonary blastoma (PPB), the most frequent primary lung malignancy in children, is highly characteristic of the syndrome. Other manifestations include ovarian Sertoli-Leydig cell tumor (SLCT), pediatric cystic nephroma, thyroid adenoma and carcinoma, embryonal rhabdomyosarcoma (ERMS) (particularly of the gynecological tract) and other rare entities[6–9].

Sarcomas are amongst the most common neoplasms in this syndrome and DICER1-associated sarcomas exhibit several characteristic morphological features, which can aid the pathologist in suspecting an association with *DICER1* PVs, irrespective of the site of origin. These features comprise a subepithelial layer of malignant mesenchymal cells (cambium layer), areas of rhabdomyoblastic differentiation with positive staining for myogenin and myoD1, cellular/immature and occasionally malignant cartilage, foci of bone/osteoid and areas of anaplasia[10–13]. Furthermore, we have recently shown that both ERMS with *DICER1* PVs and a tumor entity termed "primary intracranial sarcoma, *DICER1*-mutant" (PIS DICER1) are associated with DNA methylation signatures that are distinct from their morphological counterparts that are not DICER1-associated[14–16]. This finding, together with our recent speculations regarding the histomorphological similarities between DICER1-associated sarcomas[11–13] arising at different sites led us to question whether in general, DICER1-associated tumors share common features such that they represent a distinct tumor entity, arising at various anatomical locations.

Here, we address this hypothesis by analyzing 534 tumors, including a large number with *DICER1* PVs, by DNA methylation profiling and identify three classes of mesenchymal tumors with DICER1 alteration, comprising tumors from various anatomical locations.

## Results

### DNA methylation profiling of DICER1-associated neoplasms

We analyzed whole genome DNA methylation data of 534 tumors including various histotypes associated with the DICER1 syndrome, as well as reference entities representing morphological counterparts of the tumors studied. The study set included a total of 431 tumors with known *DICER1* mutational status (431/534, 81%) of which 176 were reported to harbor *DICER1* alterations (176/431, 41% of tumors analyzed for *DICER1* variants). Detailed information on the tumor histotypes studied and *DICER1* PVs of the study cohort is provided in Supplementary Table 1 and Supplementary Data 1.

Unsupervised hierarchical clustering and t-SNE dimensionality reduction of DNA methylation data segregated tumors into distinct and stable clusters (Fig. 1a–c and Supplementary Fig. S1). Wilms tumor (WILMS), *MYOD1*-mutant spindle cell and sclerosing rhabdomyosarcoma (SRMS), ERMS, alveolar rhabdomyosarcoma (ARMS), low-grade endometrial stromal sarcoma (LGESS), high-grade endometrial stromal sarcoma (HGESS), Müllerian adenosarcoma (MAS), embryonal tumor with multilayered rosettes (ETMR), lung adenocarcinoma (LAC), clear cell renal cell carcinoma (RCC), ciliary body medulloepithelioma (MEPL), multinodular goiter (MG) and papillary thyroid carcinoma (PCA) each formed a distinct molecular cluster defined by diagnoses based on histology and established molecular testing, irrespective of *DICER1* alteration status. For SLCT, we identified a subcluster that correlated with *DICER1* PV status, which we termed "SLCT with DICER1 alteration" (SLCT DICER1). Furthermore, we identified three closely related methylation clusters of which one mostly resembled PIS DICER1 and two represented clusters combining multiple tumor types from several anatomically distinct locations (Fig. 1d). The latter clusters represent two molecular classes that we have provisionally named "low-grade mesenchymal tumor with DICER1 alteration" (LGMT DICER1) and "sarcoma with DICER1 alteration" (SARC DICER1). As explained below, LGMT DICER1, SARC DICER1 and PIS DICER1

represent a group of mesenchymal neoplasms with DICER1 alterations, which are distinct from other DICER1-associated tumors.

Samples which did not cluster with their tumor group by institutional diagnoses (herein referred to as "outliers") were mostly tumors previously diagnosed as MAS or SLCT, which clustered among the classes referred to here as LGMT DICER1 and SARC DICER1. These cases are discussed below in detail. Furthermore, tumors for which no consensus diagnosis of uterine ERMS or MAS was made during institutional workup and central review for a prior study, segregated into clusters of MAS or SARC DICER1. Cumulative copy number profiles of all molecular classes identified by DNA methylation analysis are shown in Supplementary Fig. 2.

### Classes of mesenchymal tumors with DICER1 alteration

The three clusters of LGMT DICER1, SARC DICER1 and PIS DICER1 identified by our methylation analyses combined multiple well established clinicopathological tumor entities which are known to be highly associated with the DICER1 syndrome and which may exhibit considerable morphological overlap (Supplementary Table 2). The LGMT DICER1 class encompassed cases of cystic nephroma (CN, $n = 7$), PPB type I/Ir (PPB I/Ir; PPB I, $n = 5$ and PPB Ir, $n = 1$), PPB type II (PPB II, $n = 3$), ERMS of the uterine cervix ($n = 1$) and the fallopian tube ($n = 1$), as well as nasal chondroid mesenchymal hamartoma (NCMH, $n = 2$). The SARC DICER1 class included ERMS of various locations (uterine corpus and cervix, $n = 20$; retroperitoneum, $n = 1$; and the head and neck region, $n = 1$), PPB type II (PPB II, $n = 3$), PPB type III (PPB III, $n = 8$), anaplastic sarcoma of the kidney (ASK, $n = 3$) and pulmonary blastoma (PB, $n = 2$). The PIS DICER1 cluster almost exclusively consisted of PIS DICER1 ($n = 26$), but also encompassed one each of ASK and PPB III.

Correlation of clinical parameters with class assignment (Fig. 2a–c) revealed a significant enrichment for older ($p < 0.001$, Games-Howell post-hoc test) and female ($p = 0.037$, Chi-squared Test) patients in the SARC DICER1 class. This was due to the presence of uterine ERMS, which occurred in older patients as compared to other tumors of classes LGMT DICER1, SARC DICER1 and PIS DICER1 (Supplementary Fig. 3a, b). Furthermore, patients with kidney tumors within the SARC DICER1 class were significantly older than patients with kidney tumors of class LGMT DICER1 (median age: 8.8 years vs. 1.2 years, respectively) (Supplementary Fig. 4c). The age at diagnosis of those with lung tumors of SARC DICER1 class was greater than that of patients with LGMT DICER1 tumors in general; however, this difference was not statistically significant (Supplementary Fig. 3d).

Available survival data suggested differences between the three groups of mesenchymal tumors with DICER1 alteration, with a significantly better progression free survival for LGMT DICER1 when compared to SARC DICER1 and PIS DICER1 (Fig. 2d–f).

A comparison of clinical parameters and molecular classes is given in Table 1.

### Shared morphological features across mesenchymal tumors with DICER1 alteration

Central histopathological review (by specialist pathologists involved in the study) of 79/86 (92%) tumors which fell into clusters of LGMT DICER1, SARC DICER1 and PIS DICER1 identified morphological features shared across all three classes (Fig. 2g–r, Supplementary Fig. 4, and Supplementary Table 3). These included primitive mesenchyme with variable frequency of rhabdomyoblastic and chondroid differentiation. However, we noted a shift from predominantly cystic or glandular tumor architecture in LGMT DICER1 tumors towards a more cellular mesenchymal morphology with high-grade features and increasing frequency of anaplasia and tumor cell necrosis in high-grade tumors (SARC DICER1 and PIS DICER1).

In detail, LGMT DICER1 were predominantly cystic or glandular with no or only sparse primitive, hypocellular mesenchyme (Fig. 2g–j). In a subset of tumors, occasional rhabdomyoblast were present within

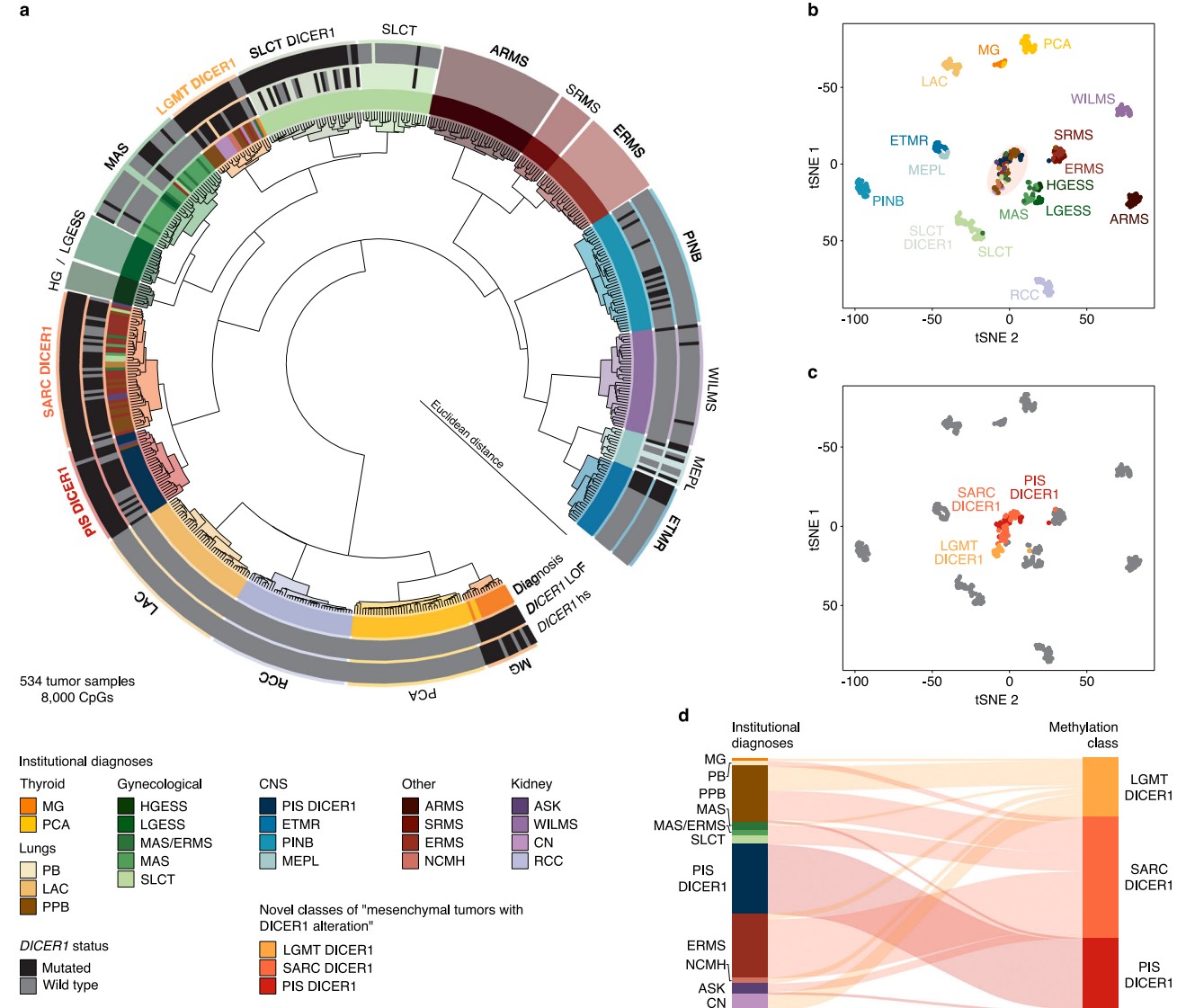

**Fig. 1 | Molecular classification of DICER1-associated neoplasms by DNA methylation analysis. a** Unsupervised hierarchical clustering (Euclidean ward) of the 8000 most differentially methylated CpGs of 534 neoplasms (related to Supplementary Fig. 1a). Samples are colored according to their institutional diagnoses. Known *DICER1* variants are denoted in black, *DICER1* wild-type alleles are indicated in dark-gray, blank annotation indicates unknown *DICER1* status. Clusters are colored according to their molecular class by DNA methylation. **b, c** 2D representation of pairwise sample correlation using the 8000 most variable methylated probes by t-SNE dimensionality reduction (related to Supplementary Fig. 1b, c). **b** Samples are colored according to their institutional diagnoses. **c** Samples are colored according to their cluster assignment of LGMT DICER1, SARC DICER1 and PIS DICER1 by unsupervised hierarchical clustering (**a**). Tumors falling into other clusters are depicted in gray. **d** Reclassification of neoplasms into three molecular classes of mesenchymal neoplasms with DICER1 alteration (LGMT DICER1, SARC DICER1 and PIS DICER1) corresponding to their institutional diagnoses. Institutional diagnoses and methylation clusters are depicted by colors as indicated.

the mesenchymal tumor component. Chondroid differentiation was present in cases of PPB and NCMH.

Conversely, SARC DICER1 predominantly consisted of primitive mesenchyme with variable cellularity and a mostly diffuse and patternless growth, although a fascicular growth pattern was present in some tumors (Fig. 2k–n). The mesenchymal tumor cells were mostly small and only mildly atypical but anaplasia and prominent rhabdomyoblastic differentiation were present in a subset of tumors. In most cases with lining epithelium, a subepithelial aggregation of tumor cells (cambium-like layer) was present.

PIS DICER1 were purely mesenchymal and were markedly cellular with a diffuse and/or fascicular growth pattern (Fig. 2o, p). Most tumors showed rhabdomyoblastic differentiation, anaplasia and tumor cell necrosis. In a subset of tumors, chondroid differentiation was present. The one PPB III assigned to the PIS DICER1 cluster showed a pure mesenchymal phenotype with a diffuse and fascicular growth

pattern, focal necrosis, and prominent anaplasia (Fig. 2q, r). Unfortunately, no hematoxylin & eosin-stained slides were available to review for the ASK falling into the PIS DICER1 cluster.

## Genetic alterations of mesenchymal tumors with DICER1 alteration

*DICER1* RNase IIIb hotspot PVs were identified in 83/86 (97%) of analyzed LGMT DICER1, SARC DICER1 and PIS DICER1, which were accompanied by *DICER1* LOF PVs (nonsense [n = 22], frameshift [n = 27] or splice site PVs [n = 7], or a large gene deletion [n = 1]) in 57/83 tumors (69%). Furthermore, a second RNase IIIa/b mutation was identified in 4/83 (5%) tumors. Of the 22 tumors in which we did not identify a second *DICER1* alteration 10/22 (46%) showed *DICER1* RNase IIIb hotspot PVs with a variant allele frequency of > 0.5, suggesting loss of heterozygosity. Of the three tumors without *DICER1* RNase IIIa/b hotspot PVs, *DICER1* frameshift variants were present in 2/3 (66%) samples.

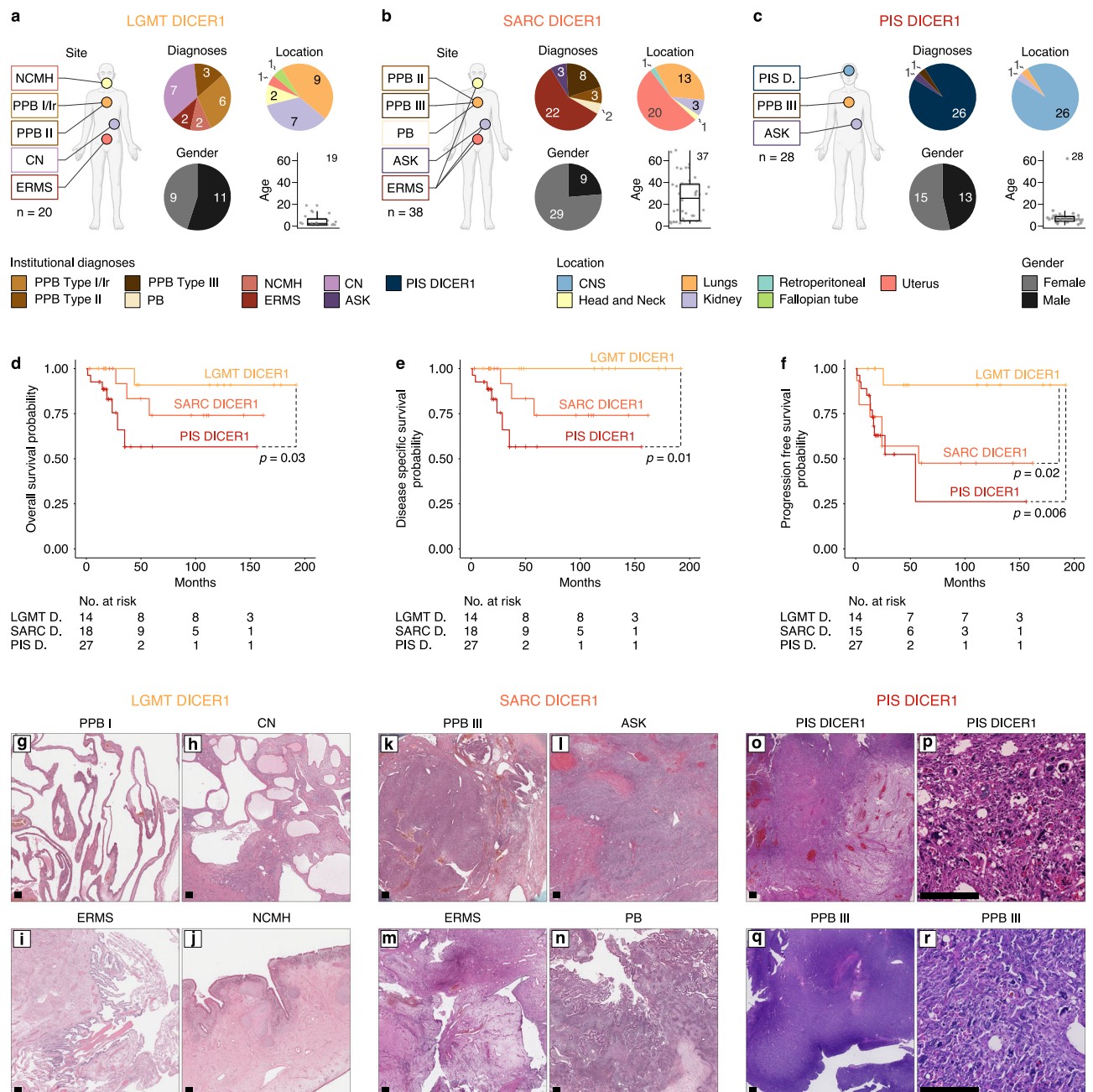

**Fig. 2 | Clinicopathological features of mesenchymal neoplasms with DICER1 alteration.** Institutional diagnoses, tumor location, gender distribution and patient age of **a** LGMT DICER1, **b** SARC DICER1 and **c** PIS DICER1 (related to Supplementary Fig. 3). Black lines mark the median, edges of boxes denote interquartile range (IQR), and vertical lines indicate 1.5 × IQR. The number of independent samples is indicated. Kaplan–Meier survival curves (Kaplan–Meier estimates) indicating (**d**) overall survival, (**e**) disease specific survival and (**e**) progression free survival for LGMT DICER1, SARC DICER1 and PIS DICER1. Significant differences are indicated by dashed lines. **g–r**, Representative histological characteristics of LGMT DICER1 (**g–j**) showing predominantly cystic or glandular configuration with no or only sparse adjacent primitive, hypo-cellular mesenchyme, SARC DICER1 (**k–n**) predominantly consisting of atypical mesenchyme with variable cellularity and a mostly diffuse and patternless growth, as well as PIS DICER1 (**o–r**) displaying a purely mesenchymal phenotype with a markedly increased cellularity and signs of anaplasia and coagulative necrosis in a subset of tumors (related to Supplementary Fig. 4 and Supplementary Table 3). Hematoxylin & Eosin, scale bar equals 250 μm.

Germline information was available for 53/86 (62%) patients, of which 28/53 (53%) showed a germline provenance of the *DICER1* LOF PV identified in the tumor and therefore these tumors arose in the context of DICER1 syndrome. An overview of identified *DICER1* PVs is provided in Fig. 3a–c (Supplementary Data 2).

Next, we performed panel-based next generation sequencing on a total of 80 tumors from all three classes (Fig. 3f, Supplementary Data 3, and Supplementary Data 4). The other genes most frequently affected were *TP53* (32/80, 40%), *KRAS/NRAS* (17/80, 21% and 6/80, 8% respectively), *KMT2D* (16/80, 20%), and *NF1* (8/80, 10%). *TP53, KRAS/NRAS,*

and *NF1* PVs were only observed in SARC DICER1 and PIS DICER1, underlining shared oncogenic mechanisms of these neoplasms.

For each of the classes, we investigated copy-number profiles generated from DNA methylation array data. LGMT DICER1 mostly showed balanced copy-number profiles with a significantly lower median genomic index (GI; total number of segmental gains or losses[2]/ number of involved chromosomes) of 14.7 (range 1–56), compared to SARC DICER1 and PIS DICER1, which showed complex genomes with broad and diverse copy-number variations (CNV) and a median GI of 92.9 (range 1–2532) and 314 (range 3 – 3216), respectively (Fig. 3d and

**Table 1 | Clinical characteristics of mesenchymal tumors with DICER1 alteration according to LGMT DICER1, SARC DICER1 and PIS DICER1 cluster assignment (n = 86)**

| Molecular class | | LGMT DICER1 | SARC DICER1 | PIS DICER1 |
|---|---|---|---|---|
| Number of patients | n = 86 | 20 | 38 | 28 |
| Diagnoses | PIS DICER1 | 0 (0%) | 0 (0%) | 26 (100%) |
| | NCMH | 2 (100%) | 0 (0%) | 0 (0%) |
| | PB | 0 (0%) | 2 (100%) | 0 (0%) |
| | PPB I/Ir | 6 (100%) | 0 (0%) | 0 (0%) |
| | PPBII | 3 (50%) | 3 (50%) | 0 (0%) |
| | PPB III | 0 (0%) | 8 (89%) | 1 (11%) |
| | CN | 7 (100%) | 0 (0%) | 0 (0%) |
| | AS | 0 (0%) | 3 (75%) | 1 (25%) |
| | ERMS | 2 (8%) | 22 (92%) | 0 (0%) |
| Location | CNS | 0 (0%) | 0 (0%) | 26 (100%) |
| | Head and Neck | 2 (67%) | 1 (33%) | 0 (0%) |
| | Lungs | 9 (39%) | 13 (57%) | 1 (4%) |
| | Kidney | 7 (64%) | 3 (27%) | 1 (9%) |
| | Retroperitoneal | 0 (0%) | 1 (100%) | 0 (0%) |
| | Fallopian tube | 1 (100%) | 0 (0%) | 0 (0%) |
| | Uterus | 1 (5%) | 20 (95%) | 0 (0%) |
| Outcome | 5 year DSS | 100% | 74.1% (CI 52.6–100) | 56.6% (CI 35.0–91.4) |
| | 5 year PFS | 90.9% (CI 75.4–100) | 47.5% (CI 26.5–85.4) | 26.2% (CI 6.1–100) |
| Age | Range (years) | 0.1–18 | 2.0–69.0 | 0.8–61 |
| | Median (years) | 1.5 | 23 | 6 |
| | NA | 1 | 1 | 0 |
| Gender | Female | 9 | 29 | 15 |
| | Male | 11 | 9 | 13 |
| | Ratio (M:F) | 1.2 | 0.3 | 0.9 |

*DSS* Disease specific survival, *PFS* Progression free survival.

Supplementary Data 2). Recurrent gains of chromosome 8 were characteristic of SARC DICER1 and were also identified in a small subset of LGMT DICER1 and PIS DICER1 (Fig. 3e). High-level oncogene amplification of *FGFR1* was found in one SARC DICER1, of *MYCN* in one PIS DICER1 and of *PDGFRA* in one LGMT DICER1 and three SARC DICER1. *CDKN2A* deletions were identified in one SARC DICER1 and three PIS DICER1.

**Sarcomas with DICER1 alteration show a genome-wide hypomethylation signature**

To gain a more detailed insight into class-specific methylation patterns we analyzed overall methylation levels of LGMT DICER1, SARC DICER1 and PIS DICER1 in the context of other neoplasms. SARC DICER1 exhibited lower global DNA methylation levels as compared to LGMT DICER1, with the lowest global DNA methylation levels observed in PIS DICER1 (Fig. 4a and Supplementary Table 4). Although less pronounced, this hypomethylation signature in both SARC DICER1 and PIS DICER1 was also observed when only CpG rich regions (CpG islands, CGI), gene bodies or promoter regions were investigated (Supplementary Fig. 5a–c and Supplementary Table 4). Mean global methylation levels and the genomic index of the three tumor classes showed a weak inverse correlation (Fig. 4b). Interestingly, hypomethylation was pronounced in genomic regions with copy number gains across all LGMT DICER1, SARC DICER1 and PIS DICER1 when compared to genomic balanced regions and regions with copy number losses (Supplementary Fig. 5d, e and Supplementary Table 6). Group-wise differential methylation analyses of the three tumor classes showed that differentially methylated probes (DMPs) identified in both SARC DICER1 and PIS DICER1 were mostly hypomethylated (Fig. 4c and Supplementary Table 5) and distributed genome-wide

(Fig. 4d and Supplementary Fig. 5f, g), which again highlights a global hypomethylation phenotype for SARC DICER1 and PIS DICER1. In line with these results differentially methylated regions (DMRs) identified in SARC DICER1 and PIS DICER1 were also mostly hypomethylated (Fig. 4e). Gene-ontology analyses of gene sets overlapping with the identified DMRs showed an enrichment for GO terms associated with gene silencing by RNA/miRNA (Fig. 4f and Supplementary Fig. 5h, i). Identification of the absolute number of overlapping DMRs between classes revealed the greatest overlap between SARC DICER1 and PIS DICER1, highlighting their close epigenetic relationship (Fig. 4g).

**Characteristics of methylation outliers**

By applying unsupervised clustering to our methylation data set we identified outliers, which did not cluster with their tumor group by institutional diagnoses, including one ERMS with *DICER1* PVs that was molecularly classified as MAS. Furthermore, there were nine tumors, initially diagnosed as MAS (n = 2), MAS/ERMS (cases which could not be confidently diagnosed as either of these tumor types due to morphological overlap) (n = 3), SLCT (n = 3), or MG (n = 1), which clustered stably with LGMT DICER1 or SARC DICER1 (Fig. 5a). In 7/9 outliers (78%) *DICER1* RNase IIIb hotspot PVs were identified, which were accompanied by *DICER1* LOF PVs in 5/7 tumors (71%). Outliers in cluster SARC DICER1 showed frequent gains of chromosome 8, analogous to other SARC DICER1 tumors. In contrast, cumulative CNV profiles of MG, MAS and SLCT DICER1 showed CNV of chromosome 8 in only in a small subset of cases (Fig. 5f–h). Histologically, SARC DICER1 outliers diagnosed as MAS or MAS/ERMS showed features of biphasic epithelial and stromal neoplasms with rhabdomyoblastic differentiation noted in one out of two tumors where this was specified (Fig. 5b, c). Outliers diagnosed as SLCT

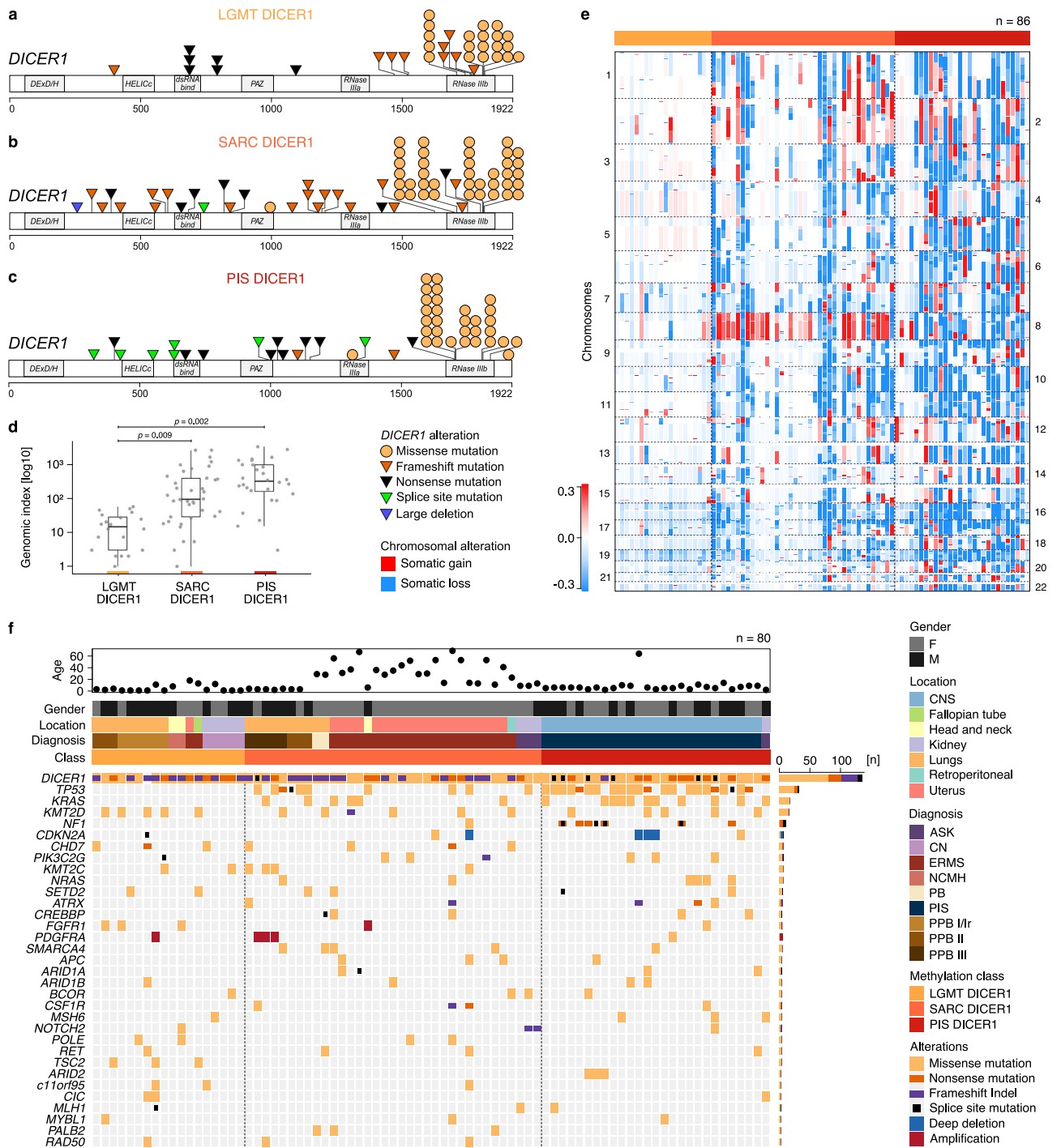

**Fig. 3 | Molecular characteristics of mesenchymal neoplasms with DICER1 alteration.** *DICER1* variants identified in **a** LGMT DICER1, **b** SARC DICER1 and **c** PIS DICER1. **d** Genomic index (total number of segmental gains or losses[2]/number of involved chromosomes) of LGMT DICER1, SARC DICER1 and PIS DICER1, indicative of genomic complexity (*n* = 86). Black lines mark the median, edges of boxes denote interquartile range (IQR), and vertical lines indicate 1.5 × IQR. Statistical significance was determined by the Games-Howell post-hoc test. **e** Case-by-case copy number profiles of LGMT DICER1, SARC DICER1 and PIS DICER1 with chromosomal gains depicted in red and losses shown in blue. **f** Variants called by panel-based DNA sequencing of 80 tumors from the three classes of mesenchymal neoplasms with DICER1 alteration. Above DNA sequencing results, patients age, gender, tumor location, institutional diagnoses and DNA methylation class assignment are annotated as indicated by the figure's legend.

revealed the frequent presence of heterologous epithelial and mesenchymal elements, including rhabdomyoblastic differentiation in 2/3 tumors (Fig. 5d, e). Although a complete morphological re-evaluation of outliers could not be achieved due to the limited availability of slides for review, these findings suggest that rhabdomyoblastic differentiation may dictate their clustering with SARC DICER1 group.

## Discussion

In this study, we identify a group of mesenchymal tumors with DICER1 alteration, which includes three classes termed LGMT DICER1, SARC DICER1 and PIS DICER1 (Fig. 6). These three tumor classes comprise mesenchymal tumors of various anatomical locations that typically harbor a combination of a *DICER1* LOF PV alongside a *DICER1* missense PV (non-classic two hit tumor suppressor PVs). LGMT DICER1 mostly

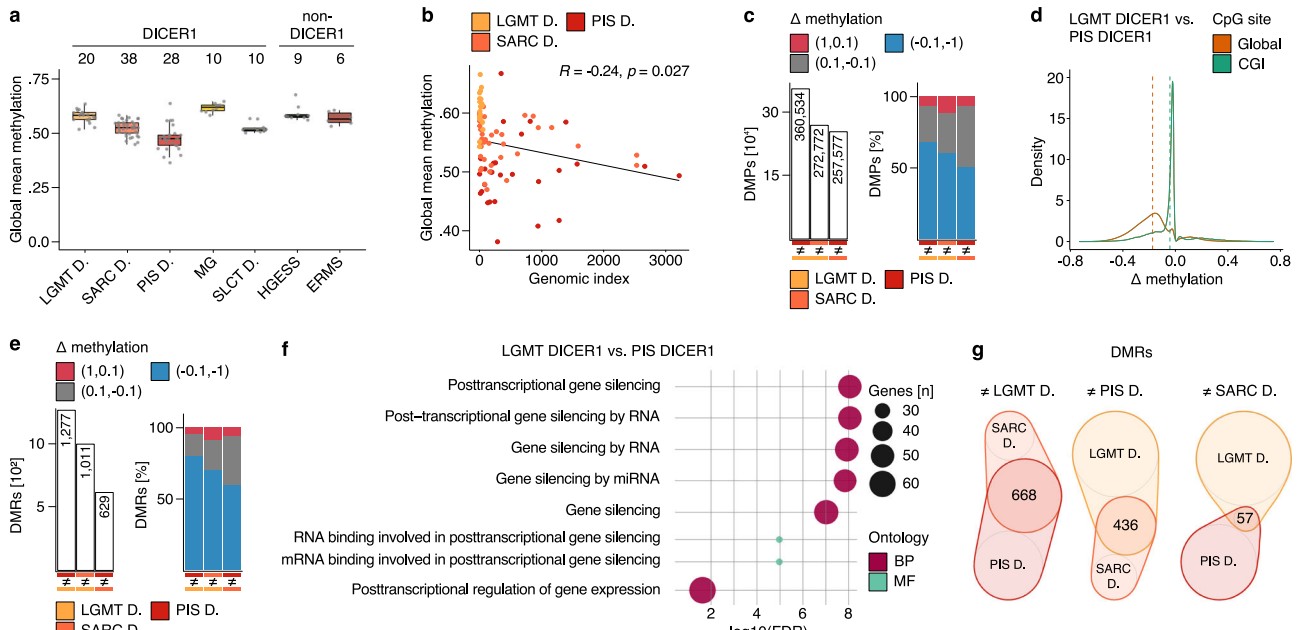

**Fig. 4 | Sarcomas with DICER1 alteration show a genome-wide hypomethylation signature. a** Global mean DNA methylation levels per sample across CpGs outside of CGIs (CpG-dense regions in CGIs are excluded for global quantifications) for LGMT DICER1, SARC DICER1 and PIS DICER1, MG and SLCT DICER1, as well as ERMS and HGESS (Supplementary Table 4). Black lines mark the median, edges of boxes denote interquartile range (IQR), and vertical lines indicate 1.5 × IQR. Number of samples analyzed, and DICER1-association are indicated at the top. **b** Global mean methylation and genomic index show an inverse correlation in LGMT DICER1, SARC DICER1 and PIS DICER1. Correlation was determined using the Pearson correlation coefficient. **c** Absolute number, and difference (Δ) in methylation of differentially methylated probes (DMPs) identified in a group-wise comparison of clusters of mesenchymal neoplasms with DICER1 alteration (LGMT DICER1, SARC DICER1 and PIS DICER1) (Supplementary Table 5). Δ in methylation indicates the difference between individual beta values: Red indicates a positive Δ in methylation

of 1 to 0.1 (hypermethylation), gray indicates a Δ in methylation between 0.1 and −0.1, and blue indicates a negative Δ in methylation of −0.1 to −1 (hypomethylation). **d** Difference (Δ) in methylation according to genome-wide distribution of DMPs identified in a group-wise comparisons between LGMT DICER1 and PIS DICER1 (related to Supplementary Fig. 5f, g). Dashed lines mark the mean difference in methylation according to genome-wide distribution of DMPs. **e** Absolute numbers, and difference (Δ) in methylation of differentially methylated regions (DMRs) identified in a group-wise comparison of LGMT DICER1, SARC DICER1 and PIS DICER1 (Supplementary Table 5). **f** Visualization of functional enrichment (gene ontology) analysis of genes overlapping with DMRs identified in a group wise comparison between LGMT DICER1 and PIS DICER1 (related to Fig. S5h, i). BP biological process, MF molecular function, FDR false discovery rate. **g** Venn diagram showing the overlap of DMRs identified in group-wise comparisons of LGMT DICER1, SARC DICER1 and PIS DICER1. Venn-diagrams are plotted to scale.

represent cystic neoplasms of various organs that usually do not harbor additional molecular alterations and show an excellent clinical outcome[17,18]. In contrast, DICER1-associated sarcomas (SARC DICER1 and PIS DICER1) usually exhibit an overtly sarcomatous phenotype which is associated with frequent alterations in *TP53, KRAS/NRAS* and *NF1*, high genomic index, as well as a global hypomethylation signature. In contrast to LGMT DICER1, DICER1-associated sarcomas may show a more aggressive clinical course[15,19–23]. Although the difference in the global methylomic profiles of these three classes may reflect their cell type composition, such as dominant epithelium component in LGMT DICER1, dominant sarcoma component in the other two classes and no epithelium in PIS DICER1, the identification of these DICER1-associated tumor classes from various anatomical sites will enable meaningful clinical trial stratification in the future and suggests that rational drug development addressing the differing molecular foundations of mesenchymal tumors with DICER1 alteration may be a plausible goal.

We also show that some other DICER1-associated lesions, such as SLCT DICER1, nodular thyroid lesions (MG and PCA), PINB, MEPL and WILMS correspond to molecular classes distinct from the classes of DICER1-associated mesenchymal tumors. This suggests that *DICER1* alterations may induce or contribute to distinct tumor phenotypes dependent on the specific cellular context. While DICER1-associated mesenchymal tumor classes may share similar cellular backgrounds, other DICER1-associated neoplasms likely have different cellular origins depending on the tumor location. It remains unclear if the subclustering in DICER1-associated sarcomas into classes of SARC DICER1

and PIS DICER1 is due to a different cell of origin for CNS and non-CNS tumors, or if site-specific factors shaping the tumor microenvironment have influenced the molecular features.

A diagnosis of mesenchymal tumor with DICER1 alteration should always be taken into consideration when dealing with a mesenchymal neoplasm with areas of a subepithelial layer of malignant mesenchymal cells, areas of rhabdomyoblastic differentiation, cellular/immature and sometimes overtly malignant cartilage or foci of bone/osteoid. The presence of an overtly sarcomatous differentiation (LGMT DICER1 vs. SARC DICER1), anatomical location (SARC DICER1 vs. PIS DICER1) and the identification of a DICER1 hotspot PV, will usually be sufficient to classify DICER1-associated mesenchymal neoplasms. However, in some cases clinical presentation, histopathology and DICER1 hotspot sequencing could yield inconclusive or contradicting results and present a diagnostic challenge for pathologists, as is frequently the case in DICER1-associated tumors of the gynecologic tract[24]. In our study, DNA methylation profiling was able to classify tumors previously diagnosed as MAS, ERMS or MAS/ERMS (cases which could not be confidently diagnosed as either of these tumor types due to morphological overlap) into specific clusters of MAS or SARC DICER1, highlighting the potential for molecular markers to aid tumor classification. Therefore, ancillary tests, such as DNA methylation profiling (in ensemble with the Heidelberg Sarcoma classifier[25]), CNV profiling or panel-based DNA sequencing may be helpful in correctly classifying a neoplasm as either one of the three classes of mesenchymal tumors with DICER1 alteration, or to exclude other DICER1-associated tumor entities, such as SLCT DICER1 or uterine MAS. A diagnostic algorithm for mesenchymal

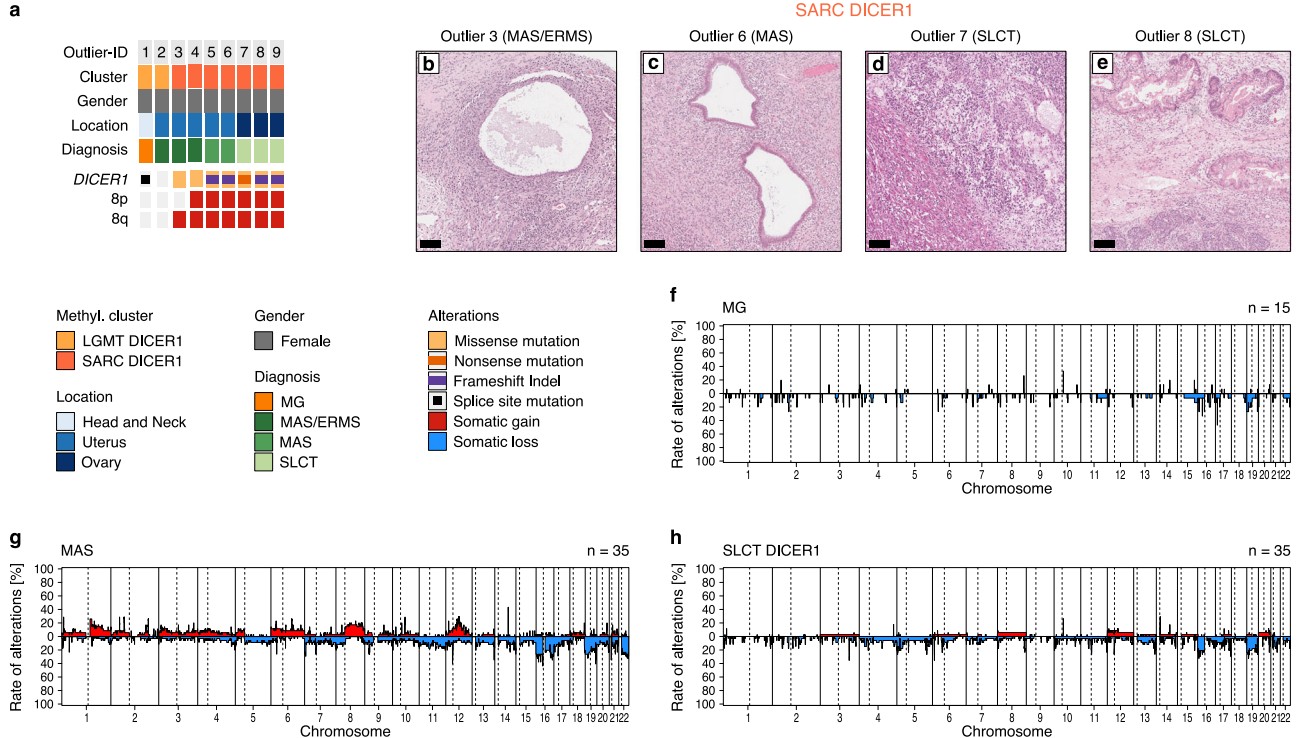

**Fig. 5 | Characteristics of DNA methylation outlier in clusters of mesenchymal neoplasms with DICER1 alteration. a** *DICER1* variants and chromosome 8 copy number status of 9 thyroid, uterine and ovarian tumors, that clustered with LGMT DICER1 or SARC DICER1 in DNA methylation analysis (Fig. 1a). Above outlier ID, DNA methylation class assignment, gender, tumor location and institutional diagnoses are annotated as indicated by the figure's legend. **b−e** Limited histomorphological evaluation of individual outliers diagnosed as MAS, MAS/ERMS or SLCT revealed (**d**) rhabdomyosarcomatous differentiation in a subset (*n* = 2) of tumors. Hematoxylin & Eosin, scale bar equals 100 μm. Cumulative copy number profiles of molecular classes of **f** MG, **g** MAS and **h** SLCT DICER1 showing the frequency of any chromosomal aberration at the respective loci (related to Supplementary Fig. 2). Chromosomal gains are depicted in red, and losses shown in blue.

tumors with DICER1 alteration is proposed in Fig. 7. Importantly, diagnosis of a mesenchymal tumor with DICER1 alteration should always prompt germline testing for DICER1 syndrome.

We consider that it would be desirable to implement the herein proposed nomenclature for mesenchymal tumors with DICER1 alteration. Nevertheless, we understand that in practice, nomenclature changes have the potential to result in confusion for clinicians and pathologists. Given that it may not be apparent that these "new" terms refer to neoplasms that already have established names in the literature rather than newly defined tumor types, this could potentially cause "loss" of important information when new terms are used. Adoption of our proposed nomenclature will require close cooperation between pathologists and clinicians dealing with these rare tumor types. Going forward, it may be rational to initially use both "new" and "old" terminology, for example "pulmonary sarcoma with DICER1 alteration (Pleuropulmonary blastoma type III)" or "cervical sarcoma with DICER1 alteration (cervical embryonal rhabdomyosarcoma)".

It is tempting to suggest that the development of DICER1-associated mesenchymal tumors at a specific anatomical location represents a disease continuum with transition from cystic to sarcomatous configuration. This hypothesis is based on the description of few patients in which a diagnosis of PPB I or CN was followed by occurrence of PPB II/III or ASK, respectively[18,26–28]. Our results support this hypothesis by showing, for example, that PPB II is found in methylation clusters of both LGMT DICER1 and SARC DICER1, whereas PPB I is only found in LGMT DICER1 and PPB III is only found in SARC DICER1 or PIS DICER1. While disruption of DICER1 has been shown to lead to downregulation of miRNAs and enhance stemness and epithelial-to-mesenchymal transition in colorectal cancer cells[29], as stated above, additional PVs in oncogenes and tumor suppressors

other than *DICER1* PVs, may contribute to this transition from a cystic to a mesenchymal phenotype.

Our study furthermore implicates global hypomethylation as a feature of sarcomatous differentiation in SARC DICER1 and PIS DICER1. Studies of Wilms tumor and glioblastoma, as well as their proposed cells of origin, have indicated that altered DNA methylation in tumors does not reflect methylation states of precursor cells, but rather represents demethylation during tumorigenesis[30,31]. Similarly, we hypothesize that that the global hypomethylation signature identified in SARC DICER1 and PIS DICER1 is result of a continuing process that persists throughout tumor progression/sarcomatous differentiation, rather than selection for a precursor cell with a pre-existing hypomethylated state. Nevertheless, further studies into the cellular origins of mesenchymal tumors with DICER1 alteration are needed to expand on our hypothesis.

When compared to LGMT DICER1, the DMRs identified in both SARC DICER1 and PIS DICER1 were widely hypomethylated and enriched for genes associated with gene silencing through miRNAs. Growing evidence indicates that disruption of miRNAs signaling may be involved in the control of DNA methylation by targeting the DNA methylation machinery[32]. In this context, our results raise the question of a potential link between disruption of the DICER1-associated miRNA machinery and global DNA methylation changes that warrants further investigation. In addition, there are other factors to consider that may contribute to demethylation. Characterization of DNA methylation associated proteins, as well as chromatin structure and composition may allow a better understanding on how DNA hypomethylation may be induced in DICER1-associated neoplasms.

Previous studies have demonstrated an association of hypomethylation and chromosomal instability in human and murine cancers[33,34]. Our results similarly show that in mesenchymal

| Methylation class | | LGMT DICER1 | SARC DICER1 | PIS DICER1 |
|---|---|---|---|---|
| Proposed class name | | Low-grade mesenchymal tumor with DICER1 alteration | Sarcoma with DICER1 alteration | Primary intracranial sarcoma with DICER1 alteration |
| Entities | | PPB I/Ir/II, CN, NCMH, ERMS | PPB II/III, ASK, ERMS, PB | PIS DICER1 |
| Characteristic alteration | | *DICER1* RNase IIIa/b hotspot missense variant + *DICER1* LOF variant (frameshift, nonsense or splice site mutation, gene deletion or LOH) | | |
| Age | Years (Density) | | | |
| Anatomical locations | CNS, Head & neck, Lungs, Kidney, Gynecological | | | |
| Morphological features | Phenotype | Cystic | Solid, botryoid | Solid |
| | Stromal cellularity | Scant | Cellular, usually moderate, may be non-uniform or diffuse | Cellular, usually marked and diffuse |
| | Stromal atypia | Usually absent | Mild to moderate, may be marked | Moderate to marked |
| Recurrent alterations | | – | Chr 8 gain, *TP53, KRAS/NRAS* | *TP53, KRAS/NRAS, NF1* |
| Genomic complexity | Genomic index | low | high | high |
| DNA methylation | Global | ~ | ↓↓ | ↓↓↓ |
| Outcome | Disease specific survival (Years) | Favorable (5 year DSS: 100%) | Intermediate (5 year DSS: 74.1%) | Unfavorable (5 year DSS: 56.6%) |

**Fig. 6 | Summary of clinicopathological and molecular characteristics of classes of mesenchymal neoplasms with DICER1 alteration.** LOF Loss of function, LOH Loss of heterozygosity.

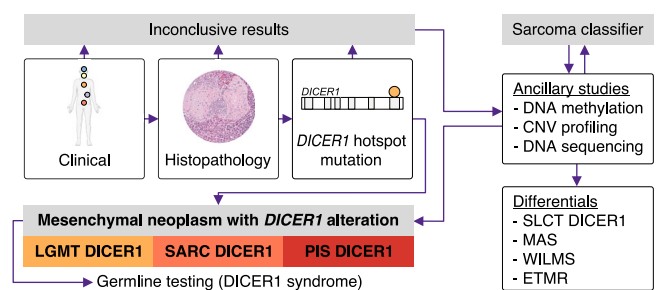

**Fig. 7 | Proposed diagnostic algorithm for mesenchymal tumors with DICER1 alteration.** Consideration of clinical features and histopathology together with the identification of a *DICER1* hotspot pathogenic variant is usually sufficient to classify DICER1-associated mesenchymal neoplasms. However, in diagnostically challenging cases ancillary molecular studies may aid tumor classification. Diagnosis of a mesenchymal tumor with DICER1 alteration should always prompt germline testing for DICER1 syndrome.

tumors with DICER1 alterations DNA hypomethylation is correlated with chromosomal instability and, more specifically, that regions affected by chromosomal gains may especially be prone to be hypomethylated.

In conclusion, we demonstrate that mesenchymal tumors with DICER1 alteration comprise three distinct clinicopathological and molecular tumor classes, which may warrant a change in nomenclature. Our study provides a combined approach to classify DICER1-associated neoplasms, which not only has diagnostic implications but will also facilitate future investigations into prognostication and therapeutic approaches for patients with such tumors.

## Methods
### Study series
This study was approved by the institutional ethics committee (Heidelberg University, S660/2020) and performed in accordance with the Declaration of Helsinki. Tumor samples of AS, CN, ERMS, HGESS, LGESS, MAS, MAS/ERMS, MEPL, MG, NCH, PB, PCA, PINB, PIS DICER1, PPB and SLCT had all undergone histopathological evaluation by specialized pathologists, as well as molecular testing whenever applicable. Samples were collected at McGill University-affiliated hospitals in Montreal, Quebec, Canada and the Institute of Pathology, University Hospital Heidelberg, mainly from collaborating institutions in accordance with ethics review board regulations. Eligible participants signed an informed consent form. Clinical data, histological features and/or sequencing results of 403/534 tumors have previously been reported elsewhere in part[14–16,21,24,27,35–64]. Wherever available, H&E-stained formalin-fixed paraffin-embedded (FFPE) sections from tumors samples were reviewed by pathologists involved in the study (FKFK, UF, MG, BAC, BCG, CSBC, FK, US, CK, WGM, AvD). Clinical data were provided by collaborators. Raw DNA methylation and

clinical data for SRMS and ARMS, as well as subsets of ERMS, LGESS and HGESS were downloaded from the DNA methylation sarcoma classifier[25] (GEO accession number: GSE140686) for a subset of PINB and ETMR from the DNA methylation CNS classifier[65] and Lambo et al. [66]. (GEO accession numbers: GSE109381 and GSE122038), for LCA and a subset of PCA from the TCGA cohorts[67,68] (https://tcga-data.nci.nih.gov/docs/publications/luad_2014/ and https://tcga-data.nci.nih.gov/docs/publications/thca_2014/), and for WILMS through the TARGET Data Matrix[69] (https://ocg.cancer.gov/programs/target/data-matrix). An overview of all tumor samples included in various analyses is provided in Supplementary Data 1.

### Statistical analyses

Statistical analyses were performed in R v.4.1.2, (Vienna, Austria) using the package *rstatix*[70]. Comparison of categorical data was performed using the Chi-squared Test. Differences between group averages were evaluated using one-way analysis of variance (ANOVA) with the Games–Howell post hoc test. All tests were two-sided. *P*-values were rounded to three decimal places. *P*-values less than 0.001 were stated as $p < 0.001$. A *p*-value of <0.05 was considered statistically significant. *P*-values of multiple comparisons were adjusted using the Bonferroni correction. Correlation was calculated using the Pearson correlation coefficient. Survival analyses (Kaplan–Meier estimates) were performed using the R-packages *survival*[71] and *survminer*[72].

### DNA extraction and array-based analysis

Genomic DNA was extracted from either fresh-frozen or FFPE tumor tissue. The Maxwell® 16 FFPE Plus LEV DNA Kit or the Maxwell 16 Tissue DNA Purification Kit (for frozen tissue) was applied on the automated Maxwell device (Promega, Madison, WI, USA) according to the manufacturer's instructions. A minimum of 100 ng DNA was extracted in every case and provided for subsequent array-based DNA methylation analyses and DNA sequencing.

### Array-based DNA methylation profiling

DNA was subject to bisulfite conversion and processed on the Infinium HumanMethylation450K BeadChip or the Illumina Infinium EPIC (850k) BeadChip (Illumina, San Diego, USA) according to the manufacturer's instructions.

### DNA methylation data pre-processing and analyses

Array data analysis was performed using R v.4.1.2, (Vienna, Austria). Data normalization and preprocessing was performed using R packages *minfi* and *Champ*[73]. For clustering analyses, Illumina450k and 850k sample data were merged into a combined dataset by selecting the intersection of probes present on both arrays (combineArrays function, *minfi*). DNA methylation data were normalized by applying background correction and dye bias correction. Probes targeting sex chromosomes, probes containing multiple single nucleotide polymorphisms, multiple hit probes and those which could not be uniquely mapped were removed. For unsupervised hierarchical clustering of DNA methylation data, 8000 and 20,000 probes with the most variably methylated probes across the dataset were selected. Distance between samples was calculated using Euclidean distance and average linkage was used to generate dendrograms. For unsupervised 2D representation of pairwise sample correlations, t-SNE dimensionality reduction was performed using the same distance metrics and default parameters. For group-wise comparisons, differentially methylated positions (DMPs) and regions (DMRs) were identified using *Champ* and the *Probelasso* method[74,75]. Only DMPs with a false-discovery rate (FDR) < 0.05 were considered. DMRs required a minimum of 5 significant probes per lasso (adjPvalProbe <0.05). DMRs were only considered to be intersecting between classes if the signs (+/−) of difference in beta values were matching.

### Copy number analysis

Copy number alterations of genomic segments were calculated from the methylation array data and plotted using the R-packages *copynumber*[76]. Summary copy number profiles were created using the R-package *conumee*[77]. Gene amplifications and deletions were identified by manual inspection. The genomic index was computed from segmented copy number data as previously described (Genomic index = total number of segmental gains and losses$^2$/number of involved chromosomes)[78]. The upper and lower thresholds for segmental gains and losses were set at 0.1 and −0.1 (log2), respectively.

### DNA sequencing and variant calling

DNA was sequenced using a customized SureSelect XT technology (Agilent) panel covering the coding regions of 201 genes (Supplementary Data 3). Library preparation, quality control, sequencing on a NextSeq or HiSeq sequencer (Illumina), and data processing were performed as previously described[79]. Reads were aligned to the reference genome hg19 and variants were annotated using ANNOVAR software[80]. Missense variants were classified as stop-loss variants, polymorphisms (variants with a frequency exceeding 1% in the healthy population as well as variants described as known polymorphisms in the single nucleotide polymorphism database) or other variants that were assessed by two prediction algorithms: SIFT[81] and PolyPhen[82]. Variants considered were SNVs with a damaging prediction by at least one algorithm or an unknown prediction by both algorithms, as well as all frameshift and nonsense mutations.

### Data visualization

Data visualization was performed in R v.4.1.2, (Vienna, Austria) using R-packages *ggplot2*[83] and *ComplexHeatmap*[84]. Parts of Figs. 2a–c, 6 and 7 (Human Clipart) were created with BioRender.com.

### Reporting summary

Further information on research design is available in the Nature Portfolio Reporting Summary linked to this article.

## Data availability

The methylation data generated in this study have been deposited in the NCBI GEO database and are publicly available under accession code GSE214568. The processed targeted sequencing data are provided in Supplementary Data 4. However, participants did not agree to publicly share their raw targeted DNA sequencing data by deposition in a public repository. The raw targeted DNA sequencing data are available from the authors upon request (andreas.vondeimling@med.uni-heidelberg.de). Raw targeted DNA sequencing data will only be shared for research-related, non-commercial purposes. All requests will be reviewed and processed within one month. There is no specific time limit for how long data will be available. All other data are available in the Supplementary Information and Supplementary Data files. Source data are provided as a Source Data file. Source data are provided with this paper.

## Code availability

The code to reproduce the main methylation analyses of this study has been deposited on figshare[85].

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

## Acknowledgements

We want to thank the Microarray Unit of the Genomics and Proteomics Core Facility, German Cancer Research Center (DKFZ) for providing excellent technical support. FKFK is supported by the Physician Scientist-Program of Heidelberg University, Germany. A.S.C. is supported by a fellowship (LCF/BQ/DI21/11860051) from "la Caixa" Foundation (ID 1000010434). A.v.D. is supported by an Illumina research grant. W.D.F. is funded by the Canadian Institutes of Health Research (FDN-148390).

## Author contributions

F.K.F.K., C.K., W.G.M., A.v.D., and W.D.F. conceived the project. F.K.F.K. and W.D.F. wrote the manuscript with input from all authors. F.K.F.K., A.S.C., A.L.S., A.v.D., and W.D.F. coordinated data generation. F.K.F.K. and D.S. analyzed methylation data. F.K.F.K. analyzed clinical data. F.K.F.K. and F.S. analyzed targeted DNA sequencing data. F.K.F.K. visualized data. F.K.F.K., D.T.W.J., C.K., W.G.M., A.v.D., and W.D.F. interpreted data. F.K.F.K., U.F., M.G., B.A.C., B.C.G., C.S.B.C., F.K., U.S., C.K., W.G.M., and A.v.D. reviewed histology. F.K.F.K., E.P., D.T.W.J., L.S.H., L.A.K., U.F., M.G., B.A.C., C.R.S., B.C.G., F.K., Y.W., D.G.H., U.S., W.G.M., A.v.D., and W.D.F. provided tumors samples and meta data. A.v.D. and W.D.F. contributed equally to this manuscript. The final manuscript was reviewed and approved off by all co-authors.

## Competing interests

The authors declare no competing interests.

## Additional information

[1]Institute of Pathology, Heidelberg University Hospital, Heidelberg, Germany. [2]Department of Human Genetics, McGill University, Montreal, QC, Canada. [3]Cancer Axis, Lady Davis Institute for Medical Research, Jewish General Hospital, Montreal, QC, Canada. [4]Molecular Mechanisms and Experimental Therapy in Oncology Program (Oncobell), Bellvitge Biomedical Research Institute (IDIBELL), L'Hospitalet de Llobregat, Avinguda de la Granvia de L'Hospitalet, Barcelona, Spain. [5]Cancer Research Program, Research Institute of the McGill University Health Centre, Montreal, QC, Canada. [6]Hopp Children's Cancer Center Heidelberg (KiTZ), Heidelberg, Germany. [7]Division of Pediatric Glioma Research, German Cancer Research Center (DKFZ), Heidelberg, Germany. [8]Department of Pediatric Oncology, Hematology and Immunology, Heidelberg University Hospital, Heidelberg, Germany. [9]Department of Pathology, University Medical Centre Utrecht, Utrecht, The Netherlands. [10]Princess Máxima Center for Pediatric Oncology, Utrecht, The Netherlands. [11]Department of Pathology, Radboud University Medical Center, Nijmegen, The Netherlands. [12]Theodor-Boveri-Institute/Biocenter, Developmental Biochemistry, Würzburg University & Comprehensive Cancer Center Mainfranken, Würzburg, Germany. [13]Department of Neuropathology, Heidelberg University Hospital, Heidelberg, Germany. [14]Clinical Cooperation Unit Neuropathology, German Consortium for Translational Cancer Research (DKTK), German Cancer Research Center (DKFZ), Heidelberg, Germany. [15]Department of Pathology, University Health Network, Toronto, ON, Canada. [16]Department of Anatomical Pathology, King Edward Memorial Hospital, Subiaco, WA, Australia. [17]School for Women's and Infants' Health, University of Western Australia, Perth, WA, Australia. [18]Department of Pathology and Laboratory Medicine, University of British Columbia, Vancouver, BC, Canada. [19]Department of Molecular Oncology, British Columbia Cancer Research Institute, Vancouver, BC, Canada. [20]Institute of Pathology, Medizin Campus Bodensee, Friedrichshafen, Germany. [21]Institute of Neuropathology, University Medical Center Hamburg-Eppendorf, Hamburg, Germany. [22]Department of Pediatric Hematology and Oncology, University Medical Center Hamburg-Eppendorf, Hamburg, Germany. [23]Research Institute Children's Cancer Center Hamburg, Hamburg, Germany. [24]Department of Pathology, Belfast Health and Social Care Trust, Belfast, UK. [25]These authors contributed equally: Andreas von Deimling, William D. Foulkes. ✉e-mail: william.foulkes@mcgill.ca

