## [Peer Review File · Nature Communications]

Genomic characterization of DICER1-associated neoplasms uncovers molecular classesREVIEWER COMMENTS

Reviewer #1 (Remarks to the Author): expertise in paediatric sarcoma and DICER syndrome genomics

Thank you for your thorough analysis and presentation of methylation data and clustering for DICER1-associated mesenchymal tumors. These results inform tumor biology and provide an important tool to support diagnosis of future cases. The figures are concise and compelling.

The methylation data convincingly support a common origin and shared tumor biology across anatomic sites. You propose to create a new system of classification which would combine cases from the lungs of infants and from uteri of older women. Please include in your discussion an argument for how this classification might be implemented in practice and acknowledge some of the challenges that this may pose to oncologists and pathologists caring for these patients.

Given that the methylation profiles largely correspond to the DICER1 mutational status, how do you envision a future diagnostic algorithm which incorporates this new tool?

Can you speculate on why some of the reported cases do not have identified DICER1 alterations?

Please more clearly define your use of the term "outlier".

Reviewer #2 (Remarks to the Author): clinical expertise in DICER syndrome

Review of "Genomic characterization of DICER1-associated neoplasms uncovers novel molecular classes"

The authors performed genome-wide methylation analysis and clustering of a large group (494) of tumors, concentrating on histologies that have been associated with DICER1-associated neoplasms. The analysis identified three specific clusters that have been associated with DICER1-associated neoplasms. The analysis identified three specific clusters that have been associated with DICER1-associated neoplasms. The analysis identified three specific clusters that have been associated with DICER1-associated neoplasms. These three groups were designated "low-grade mesenchymal tumor with DICER1 alteration" (LGMT DICER1), "high-grade sarcoma with DICER1 alteration" (HGS DICER1), and primary intracranial sarcoma, DICER1-mutant (PIS DICER1). The authors investigated other genetic events in the tumors including somatic mutations and copy number changes, as well as changes in global methylation status. Overall they suggest that this new classification scheme will improve clinical management with respect to prognostication and therapy.

This is well-designed and well-written study. The figures and tables are appropriate. The summary table contrasting the three subgroups is particularly useful. There are some areas in which further clarification would increase the impact of the manuscript.

1. The study identifies hypomethylation as a feature of the new classes of DICER1-related neoplasms. Were DNA methyltransferases/DNA demethylation genes included on the targeted NGS panel? (They seem not to be included in table S6). Can the authors comment on possible mechanisms of methylation changes—i.e. are these thought to be pre-existing features of the cells and tissue lineages that give rise to the different classes of DICER1-related tumors? Or is the hypomethylation a consequence of the mutated DICER1?
2. Please include a figure demonstrating CPG hypomethylation status in promoter vs gene body in LGMT DICER1, HGS DICER1, and PIS DICER1. These data may suggest where demethylation is "increasing" in specific genomic regions.
3. Is there clinical information regarding LGMT vs HGS vs PIS regarding tumor recurrence? We note this information is embedded in 5 year PFS. Methylation stratification could suggest groups that are more prone to recurrence.
4. Can the authors comment whether individuals with germline (mosaic) DICER1 "hotspot"

mutations more likely to have a characteristic methylation signature (e.g. more likely to be in LGMT DICER1 or HGS DICER1 or PIS DICER1).

5. Since the copy number data are derived from the genomic methylation analysis, and thus presumably yield allele-specific information, there may be an interesting opportunity to determine the relative timing of copy-number vs methylation changes during the course of tumor development. How uniform is the methylation signature at sites with copy-number gain?

6. The analysis in figure 4F is quite interesting. It would help the clarity of the presentation to specify in the text and in the figure itself (and not only in the figure legend) that the groupwise comparison is between LGMT and PIS groups. Given the roles of the DICER1 pathway in gene silencing, the enrichment of these GO terms is not too surprising. However, since both groups (LGMT and PIS) have lesions in DICER1, how do the authors account for the differential enrichment in DICER1-related processes in the two groups?

7. The authors make a convincing case that there are molecular subclasses of DICER1-related tumors in which the group identify is shared by tumors with different apparent histologic diagnoses. However at the same time, in the original clustering analysis, there are some striking examples (such as ETMR and one of the two major SLCT) in which the methylation class "trumps" the DICER1 status, i.e. tumors are clustered into a single methylation group regardless of DICER1 mutation status. How do the authors explain this result?

Minor points:

1. The different shades of green color used in figure 2a-c are difficult to distinguish
2. For non-specialists, the authors should briefly explain the significance of the number ranges ("1, 0.1"; "-0.1,-1") in figure 4c.

Reviewer #3 (Remarks to the Author): expertise in multi-omics bioinformatics and DNA methylation analysis

This manuscript entitled, " Genomic characterization of DICER1-associated neoplasms uncovers novel molecular classes," describes a new classification strategy for DICER1-associated neoplasms across 7 anatomical locations by using DNA methylation profiles and thereby enables the identification of new classes.

The authors extended their previous findings, embryonal rhabdomyosarcoma (ERMS) with DICER1 pathogenic variants (PVs) is a novel tumor type named "primary intracranial sarcoma, DICER1-mutant" (PIS-DICER1), to analyze 494 tumors with DICER1 PVs and DNA methylation profiling data to enable new mesenchymal tumor classification. Unsupervised hierarchical clustering analysis of DNA methylation profiles distinguished tumors and three classes are identified including LGMT, HGS, and PIS-DICER1. The panel-based next-generation sequencing tumors across three subtypes are performed and genetic alterations accompanied by DICER1 alteration are confirmed. A genome-wide hypomethylation signature of sarcomas with DICER1 alteration is identified. Survival analysis is then performed and showed that LGMT tumors are significantly better progression-free survival. Finally, the tumors (outliers) that are not included in the clustering analysis were characterized.

The authors are trying to address an interesting question of whether DICER1 tumors can be further divided and characterized using DNA methylation. The strategy described here is potentially useful for clinical applications. However, the findings reported have several limitations that need to be addressed as I highlight below and some conclusions are unclear what is being shown, particularly regarding the relationship between DICER1 mutation and DNA methylation.

- It is not clear how DNA methylation is relevant to DICER1 mutation since DICER1 is most important to miRNA-mediated regulation of message RNA. The authors showed that DNA methylation can distinguish tumors and further classify DICER1 tumors into 3 classes based on the unsupervised clustering analysis. However, this could be expected since DNA methylation has been used for tumor classification in a range of studies. I, therefore, suspect this classification strategy is not DICER1 mutant specific. The author should analyze if adding type-matched tumors without DICER1 mutant (which can be obtained from TCGA or other publicly available resources) can also be classified by using DNA methylation. In the meantime, the authors should demonstrate how DICER1 mutation and accompanied key mutations affect DNA transferase enzyme activity or methylation-relevant regulators in different classes.
 - There are 291 tumors associated with the DICER1 syndrome included for classification, of which 176 (60%) are DICER1 alteration tumors. It is not clear how the tumors without DICER1 mutation are classified into three categories.
 - The manuscript is overall descriptive and how these new classifications add value to our current understanding of DICER1 mutation tumors should be well discussed.
- Figure 1a is useful to understand data structure and DICER1 mutation status in different tumors. While it is difficult to assess given its current form. For example, the mutation status is not consistently illustrated using the same color scheme. Why is a different number of layers in different tumors (2 for MAS, 1 for LGESS, and 4 for HGS)? A tSNE plot for tumors colored by predicted classes is helpful to understand how tumor classes locate in tumor types.
- The author previously analyzed primary intracranial sarcoma to represent PIS-DICER1 tumors. In this study, more samples and tumor types are included, Are there any new findings for this subtype? It will be important to evaluate the molecular characteristics of PIS-DICER1 tumors newly identified from broader tumor types.
 - Nearly all the PIS-DICER1 tumors are CNS, most HGS subtype tumors are from the uterus and lung and most LGMT tumors are from the lung and kidney. Again, the classification is tumor type/location specific. The authors should raise the sample size to ensure an unbiased clustering analysis. Given that both sample sizes of lung tumors classified into LGMT and HGS are around 10, the comparison of lung tumors between these two subtypes should be performed.

RESPONSE TO REVIEWERS' COMMENTS

Reviewer #1

Thank you for your thorough analysis and presentation of methylation data and clustering for DICER1-associated mesenchymal tumors. These results inform tumor biology and provide an important tool to support diagnosis of future cases. The figures are concise and compelling.

We thank the reviewer for their positive and constructive feedback, acknowledging the insight our study provides for example with respect to informing tumor biology of DICER1-associated cancers. We have expanded our discussion in the revised manuscript, as suggested by the reviewer and outlined below.

1. The methylation data convincingly support a common origin and shared tumor biology across anatomic sites. You propose to create a new system of classification which would combine cases from the lungs of infants and from uteri of older women. Please include in your discussion an argument for how this classification might be implemented in practice and acknowledge some of the challenges that this may pose to oncologists and pathologists caring for these patients.

We thank the reviewer for this very important comment. We consider that it would be desirable to implement the herein proposed nomenclature for mesenchymal tumors with DICER1 alteration. To simplify, we have now termed the novel methylation class which we previously named “high-grade sarcoma with DICER1 alteration (HGS DICER1)” just “sarcoma with DICER1 alteration (SARC DICER1)”, and we use this new term throughout both the revised manuscript and this cover letter.

We understand that in the real world, nomenclature changes have the potential to result in confusion for clinicians and pathologists. Given that it may not be apparent that these “new” terms refer to a range of neoplasms, which each are already referred to by established terms in the literature, this could potentially result in “loss” of important information when new terms are used. Moreover, differences in treatment protocols between adults and children must be considered and adoption of our proposed nomenclature will require close cooperation between pathologists and clinicians dealing with these rare tumor types. When introducing such a new terminology, it may be rational to initially use both “new” and “old” terminology, for example “pulmonary high-grade sarcoma with DICER1 alteration (Pleuropulmonary blastoma type III)” or “cervical high-grade sarcoma with DICER1 alteration (cervical embryonal rhabdomyosarcoma)”. We have complemented our discussion with a paragraph on this topic.

2. Given that the methylation profiles largely correspond to the DICER1 mutational status, how do you envision a future diagnostic algorithm which incorporates this new tool?

When diagnosing a mesenchymal neoplasm with areas of a subepithelial layer of malignant mesenchymal cells, areas of rhabdomyoblastic differentiation, cellular/immature and sometimes overtly malignant cartilage, foci of bone/osteoid a diagnosis of DICER1-associated mesenchymal neoplasms should be taken into consideration. The presence of an overtly sarcomatous differentiation (LGMT DICER1 vs. SARC DICER1), anatomical location (SARC DICER1 vs. PIS DICER1) and the identification of a DICER1 hotspot missense variant, may usually be sufficient to classify DICER1-associated mesenchymal neoplasms. However, in some cases clinical presentation, histopathology and DICER1 hotspot sequencing may yield inconclusive or contradicting results. In such cases ancillary tests, such as DNA methylation profiling (in ensemble with the Heidelberg Sarcoma classifier), CNV profiling or panel-based DNA sequencing may be helpful in correctly classifying a neoplasm as either one of the three classes of mesenchymal tumors with DICER1 alteration, or to exclude other DICER1-associated tumor entities, such as SLCT DICER1 or uterine MAS. Importantly, diagnosis of a mesenchymal tumor with DICER1 alteration should prompt germline testing for DICER1 syndrome. We have rewritten parts of and complemented our discussion with a paragraph on this topic and included a novel figure (Fig. 7) depicting a suggested diagnostic algorithm for mesenchymal tumors with DICER1 alteration.

3. Can you speculate on why some of the reported cases do not have identified DICER1 alterations?

We could only identify one such case. This case, a cystic nephroma, fell into the LGMT DICER1 methylation cluster and we did not identify any DICER1 alterations. This case had previously been analysed for DICER1 alterations using whole-exome sequencing, as well as methylation profiling. Despite further inquiry, tumour material for this case was unfortunately unavailable for further molecular testing (resequencing of DICER1). Therefore, we are unable to comment on why no DICER1 alteration was identified. Nevertheless, the tumor morphology and DNA methylation signature highly suggest that this tumor does present a cystic nephroma (LGMT DICER1).

4. Please more clearly define your use of the term "outlier".

We thank the reviewer for their feedback. The term "outlier" refers to samples that using methylation analyses did not group with the methylation cluster expected, based on the institutional diagnoses (e.g., a sample of tumor diagnosed as SLCT with DICER1 alteration is expected to fall into the cluster of SLCT DICER1). We have made sure to state more clearly what the term outlier is referring to throughout the manuscript.

Reviewer #2

Review of “Genomic characterization of DICER1-associated neoplasms uncovers novel molecular classes”.

The authors performed genome-wide methylation analysis and clustering of a large group (494) of tumors, concentrating on histologies that have been associated with DICER1-associated neoplasms. The analysis identified three specific clusters that grouped tumors in a way that was somewhat independent of histologic classifications. These three groups were designated “low-grade mesenchymal tumor with DICER1 alteration” (LGMT DICER1), “high-grade sarcoma with DICER1 alteration” (SARC DICER1), and primary intracranial sarcoma, DICER1-mutant (PIS DICER1). The authors investigated other genetic events in the tumors including somatic mutations and copy number changes, as well as changes in global methylation status. Overall they suggest that this new classification scheme will improve clinical management with respect to prognostication and therapy.

This is well-designed and well-written study. The figures and tables are appropriate. The summary table contrasting the three subgroups is particularly useful. There are some areas in which further clarification would increase the impact of the manuscript.

We thank the reviewer for their positive comments and very helpful suggestions. Additional analyses were included in the revised manuscript, as suggested by the reviewer, and outlined below.

1. The study identifies hypomethylation as a feature of the new classes of DICER1-related neoplasms. Were DNA methyltransferases/DNA demethylation genes included on the targeted NGS panel? (They seem not to be included in table S6). Can the authors comment on possible mechanisms of methylation changes — i.e. are these thought to be pre-existing features of the cells and tissue lineages that give rise to the different classes of DICER1-related tumors? Or is the hypomethylation a consequence of the mutated DICER1?

We thank the reviewer for this very important comment, addressing potential origins for the overall methylation profiles identified in the three groups of DICER1-associated mesenchymal neoplasms.

While most tumors within the groups of DICER1-associated sarcomas (SARC DICER1 and PIS DICER1) show a relative hypomethylation signature in contrast to tumors of LGMT DICER1, there are few cases which show an overlap in mean methylation levels between all three groups. Given the above, we believe that, in contrast to the 8,000 most differentially methylated CpGs used for cluster assignment, it is most likely that the global hypomethylation signature identified in SARC DICER1 and PIS DICER1 resembles a continuing (demethylation) process that persists throughout tumor progression/sarcomatous differentiation, rather than selection for a pre-existing hypomethylated state in a precursor cell that gives rise to the tumor. This hypothesis of continuing demethylation is supported by data from studies of Wilms tumor (Ehrlich et al, Oncogene 2002) and glioblastoma (Fanelli et al, Oncogene 2008). Nevertheless, as we

have now outlined in our discussion, further studies into the cellular origins of mesenchymal tumors with DICER1 alteration are needed to expand on our hypothesis. To better support this line of thought, we have made changes to the figure depicting methylation levels of the classes (Fig. 4a and Fig. S6). Figures now show the median of the mean methylation levels per sample investigated.

Furthermore, we concur with the reviewers view that, assuming that our hypothesis of continuous demethylation is supported by other data, it would be interesting to investigate the possible mechanisms of DNA demethylation in mesenchymal tumors with DICER1 alteration, although we believe that it is likely that demethylation may have diverse causes. As suggested by the reviewer one possible explanation could be disruption of DNA methyltransferases. Unfortunately, the targeted DNA panel used in this study only includes 201 genes frequently mutated in soft tissue and CNS tumors and therefore does not cover DNA methyltransferases/DNA demethylation genes (Table S6). Nevertheless, there are other potential explanations for this difference in methylation levels. The enrichment of genes associated with gene silencing through miRNAs in mostly hypomethylated DMRs identified in SARC DICER1 and PIS DICER1 when compared to LGMT DICER1 raise the question of a potential link between disruption of the DICER1-associated miRNA machinery and global DNA methylation changes. We believe that our results will encourage and contribute to the design of further experiments aimed at understanding a potential role of DICER1 disruption in DNA methylation.

We have addressed the above points by rewriting large parts of and expanding our discussion.

2. Please include a figure demonstrating CPG hypomethylation status in promoter vs gene body in LGMT DICER1, SARC DICER1, and PIS DICER1. These data may suggest where demethylation is “increasing” in specific genomic regions.

We thank the reviewer for their suggestion. We have now included median/mean methylation levels in promoter vs. gene body in LGMT DICER1, SARC DICER1, and PIS DICER1 (Fig. S6b and c). Both gene bodies and promoter regions show a hypomethylation signature for DICER1 sarcomas.

3. Is there clinical information regarding LGMT vs SARC vs PIS regarding tumor recurrence? We note this information is embedded in 5 year PFS. Methylation stratification could suggest groups that are more prone to recurrence.

We hope that we understand the reviewers comment correctly. As the reviewer correctly states, tumors falling into the methylation groups of SARC DICER1 and PIS DICER1 are more prone to recurrences, which is reflected in 5yr PFS. More detailed information on recurrences is also given in Table S4 (Progression). Unfortunately, the rarity of

DICER1-associated neoplasms poses a challenge in terms of gathering comprehensive clinical follow-up data to provide a more comprehensive analysis of outcome.

4. Can the authors comment whether individuals with germline (mosaic) DICER1 “hotspot” mutations more likely to have a characteristic methylation signature (e.g. more likely to be in LGMT DICER1 or SARC DICER1 or PIS DICER1).

We thank the reviewer for their very interesting question. We investigated three samples (PPB I/II) of patients with mosaic DICER1 “hotspot” mutation, of which all three clustered with the group of LGMT DICER1 by methylation analysis. Furthermore, we did not identify a distinct clustering for these cases within LGMT, however these findings are limited by the small number of cases with mosaic DICER1 “hotspot” investigated. To answer, if individuals with mosaic DICER1 “hotspot” mutations are more prone to develop LGMT DICER1, rather than DICER1 sarcomas, more cases will need to be investigated in the future.

5. Since the copy number data are derived from the genomic methylation analysis, and thus presumably yield allele-specific information, there may be an interesting opportunity to determine the relative timing of copy-number vs methylation changes during the course of tumor development. How uniform is the methylation signature at sites with copy-number gain?

We thank the reviewer for their very interesting suggestion. To test whether methylation profiles observed in mesenchymal tumors with DICER1 alteration are associated with chromosomal alterations we correlated genomic index and mean global methylation levels (Fig. 4b). The results indicate that there is a correlation between increased chromosomal instability and hypomethylation, supporting our hypothesis of a continuous demethylation process in mesenchymal tumors with DICER1 alteration.

The methylation array used unfortunately does not yield information of copy number variations of alternative DNA sequences at a specified locus. Nevertheless, we believe that investigating the ratio of methylated and unmethylated probes (beta-value) in the context of segmented copy number data, as suggested by the reviewer, is a great suggestion. Our study now shows that in mesenchymal tumors with DICER1 alteration average DNA methylation levels are the lowest in genomic regions which are affected by chromosomal gains (Fig. S5d and e). Previous studies have shown that hypomethylation promotes cancer through chromosomal instability (Eden et al. Science 2003), suggesting that hypomethylation may precede chromosomal instability. Our data similarly shows that in mesenchymal tumors with DICER1 alterations DNA hypomethylation is correlated with chromosomal instability and, more specifically, that regions affected by chromosomal gains may especially be prone to be hypomethylated.

Furthermore, we have added cumulative copy number profiles of all molecular classes identified in Fig. 1a (Fig. S2).

6. The analysis in figure 4F is quite interesting. It would help the clarity of the presentation to specify in the text and in the figure itself (and not only in the figure legend) that the groupwise comparison is between LGMT and PIS groups. Given the roles of the DICER1 pathway in gene silencing, the enrichment of these GO terms is not too surprising. However, since both groups (LGMT and PIS) have lesions in DICER1, how do the authors account for the differential enrichment in DICER1-related processes in the two groups?

We thank the reviewer for their question. Figure 4f includes the groupwise comparison between LGMT DICER1 and PIS DICER1. Groupwise comparisons between LGMT DICER1 and SARC DICER1, as well as SARC DICER1 and PIS DICER1 are included in Fig. S5 g and h. To clarify we have now specified the group wise comparisons within the figures. The analyses show that there is an increasing enrichment of hypomethylated genes involved in gene silencing from LGMT DICER1 towards DICER1 sarcomas (SARC DICER1 and PIS DICER1). We believe that this may imply a continuing (demethylation) process that persists throughout tumor progression and sarcomatous differentiation. As stated by the reviewer, and in our response to comment #1 of reviewer #2, this may indicate a temporal effect (direct or indirect) of DICER1 disruption on DNA methylation. Nevertheless, there may be other effects contributing to these methylation signatures. In the light of our findings, future experiments aimed at understanding a potential effect of DICER1 disruption on DNA methylation are indicated. We have addressed the above points by rewriting large parts of and expanding our discussion.

7. The authors make a convincing case that there are molecular subclasses of DICER1-related tumors in which the group identify is shared by tumors with different apparent histologic diagnoses. However at the same time, in the original clustering analysis, there are some striking examples (such as ETMR and one of the two major SLCT) in which the methylation class “trumps” the DICER1 status, i.e. tumors are clustered into a single methylation group regardless of DICER1 mutation status. How do the authors explain this result?

We thank the reviewer for their question. We fully agree with the reviewer’s view, that methylation analysis “trumps” DICER1 status regarding tumor classification based on DNA methylation. Therefore, DNA methylation analysis could be clinically helpful, in cases where clinical, histologically and DICER1 sequencing results are inconclusive (see Fig. 7). We believe that our results indicate, that DICER1 alterations may contribute to tumorigenesis of different tumor entities in a cell of origin dependant matter, reflected in differing methylation profiles (e.g., SLCT DICER1 and SARC DICER1 may develop from differing cellular backgrounds, although both are driven by DICER1 aberrations). In the first section of results we write: “Wilms tumor (WILMS), ... each formed a distinct molecular cluster defined by diagnoses based on histology and established molecular testing, irrespective of DICER1 alteration status.”. Furthermore, in the discussion we write: “We also show that some other DICER1-associated lesions, such as SLCT DICER1, nodular thyroid lesions (MG and PCA), PINB, MEPL and WILMS correspond

to molecular classes distinct from the classes of DICER1-associated mesenchymal tumors. This suggests that DICER1 alterations may induce or contribute to distinct tumor phenotypes dependent on the specific cellular context. While DICER1-associated mesenchymal tumor classes may share similar cellular backgrounds, other DICER1-associated neoplasms likely have different cellular origins depending on the tumor location.”. We hope that this sufficiently explains the reviewer’s question.

Minor points:

8. The different shades of green color used in figure 2a-c are difficult to distinguish.
9. For non-specialists, the authors should briefly explain the significance of the number ranges (“1, 0.1”; “-0.1,-1”) in figure 4c.

We thank the reviewer for their thoughtful suggestions, and we have amended the colouring of Figure 2 and expanded Figure 4c’s legend.

Reviewer #3

This manuscript entitled, “Genomic characterization of DICER1-associated neoplasms uncovers novel molecular classes,” describes a new classification strategy for DICER1-associated neoplasms across 7 anatomical locations by using DNA methylation profiles and thereby enables the identification of new classes.

The authors extended their previous findings, embryonal rhabdomyosarcoma (ERMS) with DICER1 pathogenic variants (PVs) is a novel tumor type named “primary intracranial sarcoma, DICER1-mutant” (PIS-DICER1), to analyze 494 tumors with DICER1 PVs and DNA methylation profiling data to enable new mesenchymal tumor classification. Unsupervised hierarchical clustering analysis of DNA methylation profiles distinguished tumors and three classes are identified including LGMT, SARC, and PIS-DICER1. The panel-based next-generation sequencing tumors across three subtypes are performed and genetic alterations accompanied by DICER1 alteration are confirmed.

A genome-wide hypomethylation signature of sarcomas with DICER1 alteration is identified. Survival analysis is then performed and showed that LGMT tumors are significantly better progression-free survival. Finally, the tumors (outliers) that are not included in the clustering analysis were characterized.

The authors are trying to address an interesting question of whether DICER1 tumors can be further divided and characterized using DNA methylation. The strategy described here is potentially useful for clinical applications. However, the findings reported have several limitations that need to be addressed as I highlight below and some conclusions are unclear what is being shown, particularly regarding the relationship between DICER1 mutation and DNA methylation.

We thank the reviewer for their comments and constructive feedback. We have expanded our discussion in the revised manuscript, as suggested by the reviewer and outlined below.

1. It is not clear how DNA methylation is relevant to DICER1 mutation since DICER1 is most important to miRNA-mediated regulation of message RNA. The authors showed that DNA methylation can distinguish tumors and further classify DICER1 tumors into 3 classes based on the unsupervised clustering analysis. However, this could be expected since DNA methylation has been used for tumor classification in a range of studies. I, therefore, suspect this classification strategy is not DICER1 mutant specific. The author should analyze if adding type-matched tumors without DICER1 mutant (which can be obtained from TCGA or other publicly available resources) can also be classified by using DNA methylation. In the meantime, the authors should demonstrate how DICER1 mutation and accompanied key mutations affect DNA transferase enzyme activity or methylation-relevant regulators in different classes.

We agree with the reviewers view that the methylation-based cluster assignment is most likely driven by cellular origin, and not DICER1 mutation status. As noted by the reviewer, landmark studies by our groups and others have previously shown the value of DNA methylation based-tumour classification (Capper et al, Nature 2019 and Koelsche et al, Nature Comm 2021). The novelty of our current study lies in the identification of novel molecular tumor classes, which combine multiple well established tumor entities of various anatomical locations.

To increase the statistical validity of our study, we have increased our series of cases to 534 samples (an increase of 8%) by including more “location matched”, DICER1-wild type samples (TCGA, renal cell carcinoma of the kidney).

Nevertheless, the reviewer is raising an interesting point: Analysing the effect of DICER1 alterations on DNA transferase enzyme activity or methylation-relevant regulators may yield revealing results, given our results of global hypomethylation observed in classes of SARC DICER1 and PIS DICER1 (see response to question #1 and #6 of reviewer #2). We believe that such analyses are beyond the scope of the current study but hope that our results will contribute to the design of further experiments aimed at understanding the role of DICER1 alterations in tumorigenesis and a potential effect on DNA methylation.

2. There are 291 tumors associated with the DICER1 syndrome included for classification, of which 176 (60%) are DICER1 alteration tumors. It is not clear how the tumors without DICER1 mutation are classified into three categories.

We thank the reviewer for their comment. After expansion, our case series now includes 534 tumors of which 431 have a known DICER1 mutational status. Of the 431 tumors 176 harbour a DICER1 alteration. Classification of all 534 tumors is based on methylation clustering in Fig. 1a/Table S2. The three novel classes identified include 86 tumors, that are detailed in Fig. 2 – 4, as well as 9 outliers, as depicted in Fig. 5. All other

tumors are classified according to Fig 1a/Table S2. Details are given in the first paragraph of the results section. We hope this clarifies the matter.

3. The manuscript is overall descriptive and how these new classifications add value to our current understanding of DICER1 mutation tumors should be well discussed.

We acknowledge that our manuscript would benefit from a more detailed discussion. We have rewritten large parts of our discussion, which now included a detailed discussion on diagnostic proceedings and terminology, as detailed in our response to questions #1 and #1 by reviewer #1, as well as DNA methylation.

4. Figure 1a is useful to understand data structure and DICER1 mutation status in different tumors. While it is difficult to assess given its current form. For example, the mutation status is not consistently illustrated using the same color scheme. Why is a different number of layers in different tumors (2 for MAS, 1 for LGESS, and 4 for SARC)? A tSNE plot for tumors colored by predicted classes is helpful to understand how tumor classes locate in tumor types. **We thank the reviewer for their comment. In Fig. 1a black indicates DICER1-mutated and grey indicates DICER1 wild-type (for both hotspot and loss of function variants). If there is no information on DICER1 status, annotations are blank. To increase clarity, we have made changes to Fig. 1a and expanded the figures legend.**

5. The author previously analyzed primary intracranial sarcoma to represent PIS-DICER1 tumors. In this study, more samples and tumor types are included, Are there any new findings for this subtype? It will be important to evaluate the molecular characteristics of PIS-DICER1 tumors newly identified from broader tumor types.

We acknowledge that methylation and mutation signatures of PIS DICER1 have previously been described and the appropriate literature has been cited throughout. In the current study we show for the first time, that PIS DICER1 is closely related to other mesenchymal tumors with DICER1 alteration, a previously much discussed hypothesis. Furthermore, we show that PIS DICER1 is associated with a high genomic index and shows a global hypomethylation signature. We believe that our results, in synopsis with previous findings, significantly advances our understanding of PIS DICER1.

6. Nearly all the PIS-DICER1 tumors are CNS, most SARC subtype tumors are from the uterus and lung and most LGMT tumors are from the lung and kidney. Again, the classification is tumor type/location specific. The authors should raise the sample size to ensure an unbiased clustering analysis. Given that both sample sizes of lung tumors classified into LGMT and SARC are around 10, the comparison of lung tumors between these two subtypes should be performed.

While we understand that our small sample size poses a challenge for unbiased clustering analysis, the rarity of DICER1-associated neoplasms poses a challenge in terms of

gathering additional tumor material to expand on the analyses of these enigmatic tumor entities. However, we have added a dataset of 40 location matched samples (renal cell carcinoma of kidney) to expand the dataset. The dataset now includes location matched, DICER1-wild type samples of most investigated tumor locations (CNS, lungs, kidney, uterus, ovary and soft tissue).

We concur with the reviewer's view, that the classification is tumor class specific. However, we do not believe that the classification of LGMT DICER1 and SARC DICER1 is location specific. To elaborate on this point, we have expanded our clustering analyses. These now include clusterings of all tumor samples based on tumor location and gender to exclude potential clustering confounders (now Fig. S1 b and c). As suggested by the reviewer, we have also performed a sub-analysis of pulmonal tumors only (Rebuttal letter Fig. 1). Here, the clustering recapitulates the cluster assignment from Fig. 1a.

Rebuttal letter Fig. 1: Molecular classification of 23 pulmonary DICER1-associated neoplasms by DNA methylation analysis. Unsupervised hierarchical clustering (Euclidean ward) of the 10,000 most differentially methylated CpGs. Samples are colored according to their institutional diagnoses and DNA methylation class assignment (Fig 1a).

REVIEWERS' COMMENTS

Reviewer #2 (Remarks to the Author):

The authors have addressed the concerns of the reviewers with a very thoughtful and comprehensive response. The revised manuscript is much improved, and I have no further concerns that need addressing.

Reviewer #3 (Remarks to the Author):

The authors have addressed my concerns.